# Identifying the genes impacted by cell proliferation in proteomics and transcriptomics studies

**Marie Locard-Paulet** ⓘ**, Oana Palasca, Lars Juhl Jensen** ⓘ *

Novo Nordisk Foundation Center for Protein Research, University of Copenhagen, Denmark

* lars.juhl.jensen@cpr.ku.dk

**Data Availability Statement:** All the scripts and input tables associated to this study are available on Zenodo.org (10.5281/zenodo.6346643) under a BSD2-Clause "Simplified" license.

## Abstract

Hypothesis-free high-throughput profiling allows relative quantification of thousands of proteins or transcripts across samples and thereby identification of differentially expressed genes. It is used in many biological contexts to characterize differences between cell lines and tissues, identify drug mode of action or drivers of drug resistance, among others. Changes in gene expression can also be due to confounding factors that were not accounted for in the experimental plan, such as change in cell proliferation. We combined the analysis of 1,076 and 1,040 cell lines in five proteomics and three transcriptomics data sets to identify 157 genes that correlate with cell proliferation rates. These include actors in DNA replication and mitosis, and genes periodically expressed during the cell cycle. This signature of cell proliferation is a valuable resource when analyzing high-throughput data showing changes in proliferation across conditions. We show how to use this resource to help in interpretation of *in vitro* drug screens and tumor samples. It informs on differences of cell proliferation rates between conditions where such information is not directly available. The signature genes also highlight which hits in a screen may be due to proliferation changes; this can either contribute to biological interpretation or help focus on experiment-specific regulation events otherwise buried in the statistical analysis.

## Author summary

Nowadays, one can routinely measure how thousands of genes and proteins are regulated using so-called omics technology. This is used in many areas of biology, for example, to explore the differences between cancer cell lines and to understand what drugs do to the cells in our body. Interpreting the results of these experiments is challenging: it often results in a list of hundreds of regulated genes, which makes it difficult to pinpoint specific genes for follow up with further studies. Here, we combined data sets from two omics technologies—proteomics and transcriptomics—of more than a thousand cancer cell lines growing at different speed. We calculated the correlation of all their genes to how fast the cells were growing, to find genes that correlate reproducibly in both proteomics and transcriptomics data. These constitute what we call a "proliferation signature", which can tell us how fast the cells are growing in proteomics and transcriptomics experiments,

**Funding:** MLP, OP and LJJ are supported financially by the Novo Nordisk Foundation (Grant agreement NNF14CC0001). The funder had no role in study design, data collection and analysis, decision to publish, or preparation of the manuscript.

**Competing interests:** The authors have declared that no competing interests exist.

where this cannot be easily measured. Furthermore, these signature genes can be regulated not because of the specific treatment or disease of interest, but because of changes in cell growth that were not accounted for in the experimental plan. This resource helps target selection in screens by revealing experiment-specific regulation events, otherwise buried in a long gene list.

## Introduction

Nowadays, high-throughput proteome profiling allows relative quantification of thousands of proteins across samples. It is used in many biological contexts to characterize differences between cell lines and tissues, determine drug mode of actions, identify drivers of drug resistance, to name a few. While this reveals meaningful gene regulations across numerous conditions, these results can be confounded by secondary effects of a given treatment (or biological context). For example, a change in cell proliferation is a common undesired side effect of biological treatment and a well acknowledged confounding factor that influences results without being the intended effect of a given treatment [1]. Indeed, differences in cell growth rates correlate with the proportion of cells in each phase of the cell cycle: less proliferative cells have longer G1 or G2 phases than more proliferative cells. Consequently, slower-growing cell cultures will have more cells in G1 and G2 phase and fewer in S and M phase [2], and S and M phase-specific proteins will thus be less abundant in the lysates.

Genes highly expressed in proliferative cells have been used as proliferation markers by pathologists and researchers for many years [3–5]. Their expression indeed often correlates with the proportion of cells in S and M phase in a given sample and can strongly correlate with tumor progression and prognosis [6]. Nevertheless, there is to our knowledge no study that determines which proteins confound hypothesis-free high-throughput data analysis by correlating with cell proliferation, and the overall impact of cell growth rate on the transcriptome and the proteome remains to be determined.

In this work, we first define a pseudo-proliferation index based on transcriptomics and proteomics data for cells with known proliferation rate. We use this to analyze even larger datasets to identify a list of genes that correlate with cell proliferation at both transcript and protein level. These genes constitute a cell proliferation signature that is a valuable resource to identify and analyze datasets where proliferation is affected. We illustrate this in the context of proteomics cancer classification and drug screens [6,7], where identifying these signature genes allows to quickly discard less relevant changes that may be explained by change in cell proliferation and focus on genes that are regulated in a more context-specific manner.

## Results and discussion

### Pseudo-proliferation index derived from transcriptomics and proteomics data

Cell doubling times, or growth rates (*growth rate = ln(2)/doubling time*), are rarely provided alongside proteomics and transcriptomics data, so calculating correlation between gene relative quantities and cell growth rates is only possible for a limited number of publicly available data sets. For this reason, we defined a list of proliferation markers for which relative abundances reflect cell proliferation at protein and transcript level that would then be used to calculate an index for relative cell proliferation in datasets with no growth rates reported. The NCI60 cell lines [8] have been extensively characterized with high-throughput proteomics [9–

12] and transcriptomics [13–15] and their doubling times are publicly available from the Developmental Therapeutics Program (DTP) website (dtp.cancer.gov/discovery_development/nci-60/cell_list.htm; update of the 05/08/15). Gholami *et al.* [10] and Guo *et al.* [11] obtained pellets from DTP and lysed them directly, while Frejno *et al.* [9] obtained the cell lines from DTP and followed the DTP recommendations for *in vitro* growth. We used these data sets to identify proliferation markers that would reproducibly correlate with cell growth rates in proteomes.

We calculated the Pearson correlation with growth rates for each of the 3,645 protein groups quantified in at least two of the four NCI60 proteome data sets. Among these, we found nine human proteins that were reported as proliferation markers in the literature [3,16]. Most of these are transcribed at specific phases of the cell cycle [17] (green line in Fig 1a, and colored in S1 Fig). Although not referenced as cycling in Cyclebase v3.0, MCM3, MCM7 and MYBL2 have been shown to be expressed in a cell cycle-dependent fashion in single-cell transcriptomics [18] where MCM3/7 and MYBL2 expression peaks in G1 and G2, respectively. In the same study, CCND1 is found cyclic at protein but not transcript level, peaking in G1.

Fig 1a shows that the expression of most of these proliferation markers correlate strongly with NCI60 cell growth rates. We hypothesized that other cycling genes could be good markers of cell proliferation, and that increasing the number of genes used to estimate cell proliferation would be more robust to missing values and quantification uncertainties. Among the genes known to cycle at transcript level according to [17,19], eighteen were quantified in minimum two of the NCI60 proteomics data sets (colored points in S1a Fig). These proteins form complexes with other subunits that were not identified as cycling at RNA level but could correlate with cell growth rates; examples include the DNA polymerases A complex known to bind the cycling primases PRIM1 and PRIM2, or members of the replication factor C (RFC5 was not detected in [19] so its cycling status is unknown) (empty circles in S1a Fig).

Since we wanted to estimate relative cell proliferation in transcriptomics as well as in proteomics data, we also analyzed two transcriptomics data sets of the NCI60 cell lines grown according to the DTP recommendations [14,15] (S1b Fig). Fig 1b shows the correlation of the selected genes with cell growth rates in the transcriptome (horizontal axis) and the proteome (vertical axis). The periodic genes with the strongest correlation peak in G1/S and S phase at transcript and protein level, respectively [17]. From these data, we defined a set of potential proliferation markers containing the genes presenting high correlation with cell growth rates both in the transcriptomics and proteomics data sets (Fig 1b, grey area in the top-right corner). We compared pseudo-proliferation indexes calculated as the mean signal of:

- proliferation markers referenced in the literature (PCNA, MCM2–7, PLK1 and MKI67).

- proliferation markers referenced in the literature and genes known to cycle at transcript level (FEN1, RRM1, RRM2, CDK1, RPA2, RFC4, RFC2, PRIM2).

- all the above plus the known interacting non-cycling subunits POLA1, RFC3, RPA1, RPA3, RFC5, SMC3, STAG2, SMC1A.

We compared how the resulting pseudo-proliferation indexes correlated with cell growth rates in the proteomics NCI60 data sets (Fig 1c). As expected, the more genes were included in the proliferation markers list, the stronger the correlation. Based on these results, we decided to include the proliferation markers, periodic genes, and subunits of cycling complexes to calculate pseudo-proliferation index (all proteins in the top-right corner of Fig 1b). We performed the same comparison using the median instead of the mean of relative signals of proliferation markers. This led to lower Pearson correlations with cell growth rates and more variability between proteomics data sets (S2 Fig).

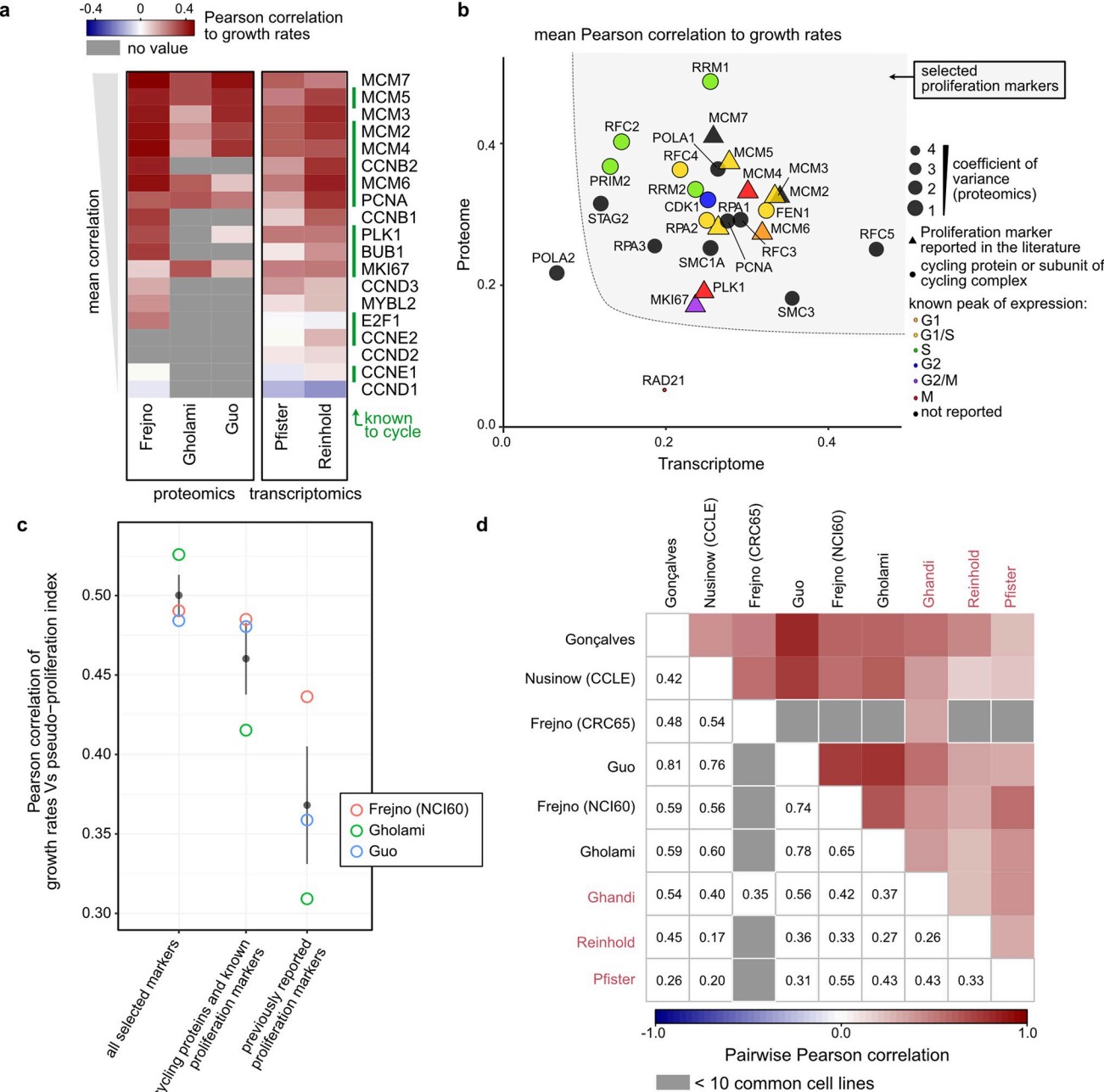

**Fig 1. Calculation of pseudo-proliferation index. a)** Pearson correlation with growth rates of the NCI60 cell lines that follow the Developmental Therapeutics Program (DTP)'s growing instructions for proliferation markers referenced in the literature. The proteins that cycle in Cyclebase 3.0 are indicated by green bars on the right ("known to cycle"), and the data set names are indicated in the bottom. **b)** Set of known markers, cycling genes and complex-associated subunits considered as proliferation marker for pseudo-proliferation index calculation. Mean Pearson correlation with growth rates in the datasets presented in (a) in the proteome (vertical axis) and transcriptome (horizontal axis). The point size is proportional to the inverse of the coefficient of variance in the proteomics data, proteins present in less than 2 data sets were excluded, as well as the cycling CDC27 due to its negative correlation with growth rates in the proteomics data sets. Periodic genes are color-coded by the phase of their expression peak, proliferation markers reported in the literature are indicated by triangles. The selected proliferation markers are indicated by the grey area. **c)** Pearson correlations between pseudo-proliferation index and growth rates in the proteomics data sets presented in (a) using the mean signal of the proliferation markers as selected in (b) (grey area), all the previously reported proliferation markers, or the previously reported proliferation markers and cycling genes with the exclusion of RAD21. Grey points and bars are mean and confidence intervals across data sets. **d)** Pairwise Pearson correlation between the pseudo-proliferation indexes calculated in the different data sets (proteomics and transcriptomics in black and red, respectively) Pairwise comparisons with less than 10 cell lines were excluded (in grey).

This data-driven approach was used to estimate relative cell proliferation on proteomics data sets with no growth rates reported: the proteomes of the CRC65 cancer cell lines [9]; the Cancer Cell Line Encyclopedia (CCLE) that comprises the CRC65, NCI60 and other cell lines [12,20]; and the recently published Pan-Cancer panel [21] (S3 Fig shows the cell lines present in each panel). For each data set, we first calculated the pseudo-proliferation indexes, and next the correlations of each protein to this proxy for cell proliferation. Gene set enrichment analysis (GSEA) showed that proteins involved in chromatin remodeling, DNA replication and chromosome organization were highly correlated to pseudo-proliferation index (S4 Fig). These results were similar to those of a GSEA performed on proteins ranked by their correlation to growth rates when available ("NCI60 only"), which confirmed that pseudo-proliferation index reflects the proliferative state of cells and can be used as an estimation of relative cell growth rates. We further controlled that the same Gene Ontology (GO) terms were reproducibly enriched across larger data sets of non-NCI60 cell lines. With the same approach, we calculated pseudo-proliferation index at RNA level in data sets containing the NCI60 and CCLE cells transcriptome [13–15] (S2 Table). Pseudo-proliferation indexes were highly consistent across proteomes (0.42 to 0.81) as well as between proteomes and transcriptomes (Fig 1d), although the growth rates of the same cell line can vary between data sets due to differences in experimental conditions and cell passages [22].

## Identification of a proliferation gene signature

Using pseudo-proliferation index, we could identify which protein quantities correlated with cell proliferation rates in the six proteomics data sets presented above (S5a Fig). We filtered out the proteins that were detected in less than two data sets and calculated the mean of Pearson correlations to pseudo-proliferation index across data sets.

We benchmarked our approach with three sets of genes expected to be highly expressed in proliferative cells (*i.e.* gold standards) either because they are known to be expressed in a cell-cycle-dependent fashion or because they were reported to be expressed under the control of a transcription factor only active on S-phase entry:

- B1: 48 genes known to be periodically expressed in synchronized cell cultures [23].

- B2: 382 genes compiled from two lists of proposed E2F transcription factor targets [19,24,25].

- Cyclebase 3.0: 570 periodically-expressed genes (https://cyclebase.org/) [17].

We ranked the proteins (excluding the proliferation markers used to calculate pseudo-proliferation index in the first place) by decreasing absolute mean of correlation to pseudo-proliferation index and counted the number of proteins belonging to each of the three gold standard sets (Fig 2a). As expected, these gold standards were enriched for the proteins most strongly correlated with cell growth rates (left of the horizontal axis). We determined a cutoff for correlation with pseudo-proliferation index: $\geq 0.344$ (Fig 2a). The exact same strategy was applied with three transcriptomic data sets to determine a transcriptomics confidence threshold of $\geq 0.567$ (Fig 2b). In both analyses, we calculated gene correlations with randomized pseudo-proliferation index (50 iterations) to check that all the gene signatures had a FDR under 0.1% (see material and methods).

Fig 2c shows gene correlations to pseudo-proliferation index at transcript and protein level. Overall, transcripts presented a higher mean correlation with pseudo-proliferation index than the proteins these were translated to, and the distribution of Pearson correlations to pseudo-proliferation index was wider at transcript than protein level. This indicates post-translational

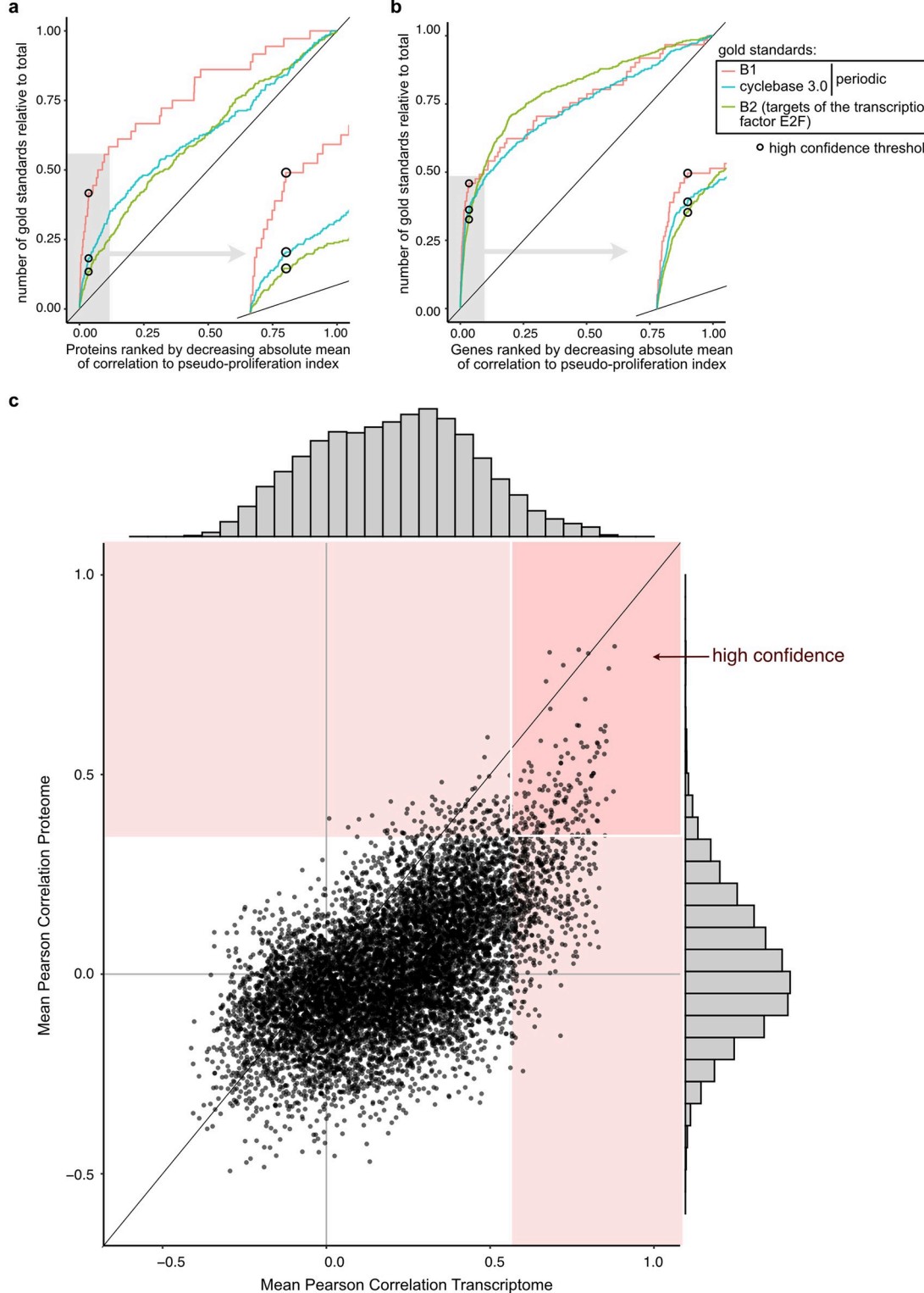

**Fig 2. Signature genes of cell proliferation. a-b)** Definition of the cutoff for correlation with pseudo-proliferation index with three sets of gold standards in the proteomes (a) and the transcriptomes (b). Proteins/genes were ranked by decreasing absolute Pearson correlation to pseudo-proliferation index (horizontal axis) and the vertical axis presents the cumulative number of gold standards for each set. Proteins/genes quantified in less than 3 and 2 data sets were excluded in (a) and (b), respectively. **c)** Scatter plot of the mean Pearson correlation to pseudo-proliferation index at protein (vertical axis) and transcript (horizontal axis) level

across all data sets. The red areas contain the proteins above the threshold in the proteome and/or transcriptome and the rectangle with white borders indicates the final list of proliferation signature genes defined in this study. The point distribution in the proteomes and transcriptomes are presented on the sides of the plot.

adjustment of protein quantities and/or post-transcriptional regulatory processes. Gene correlations to pseudo-proliferation index at protein and transcript level are available in S3 Table. We defined a threshold for a signature of cell proliferation constituted of 157 genes that correlate with pseudo-proliferation index at transcript as well as protein level.

Fig 3 shows the physical interactions between the proliferation signature genes according to the STRING physical interaction subnetwork. Each node (gene) is colored with its Pearson correlation with pseudo-proliferation index in each data set (ring). These are involved in DNA replication and mitosis. As expected, we find back all the genes used for calculating the pseudo-proliferation index (circled in black) except STAG2 and PLK1, which were just under

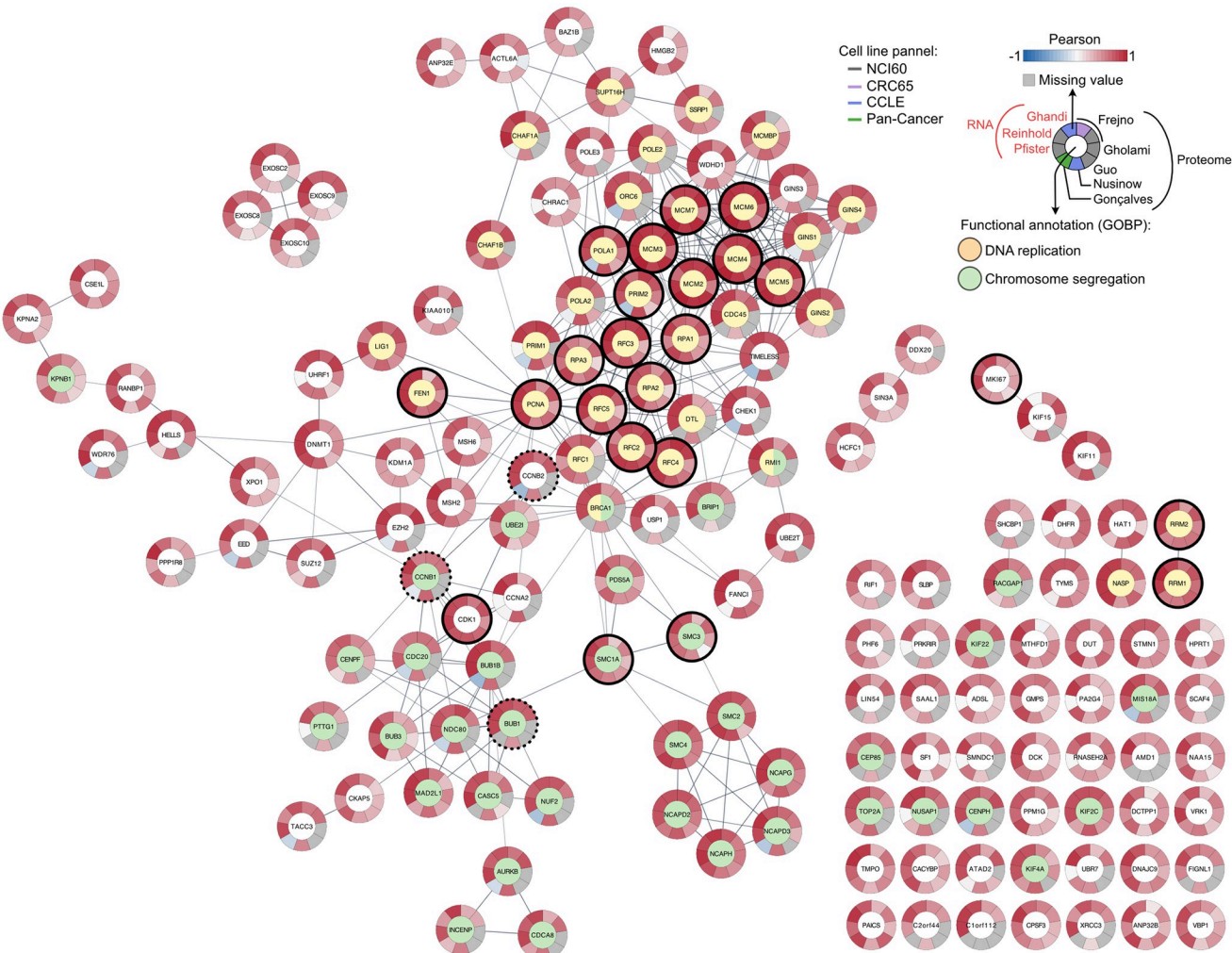

**Fig 3. Proliferation signature.** STRING subnetwork of physical interactions (score ≥ 0.7) corresponding to the proliferation signature as selected in Fig 2. The genes used to calculate the pseudo-proliferation index and the known proliferation markers not included for pseudo-proliferation index calculation are highlighted by black solid and dashed borders, respectively. The nodes are color-coded by selected gene annotations of biological processes. External ring are the Pearson correlations for each data set independently.

the threshold in the proteomics data (and the transcriptomics data for STAG2). Some of the proliferation markers previously described in the literature were also found in the proliferation signature, such as BUB1 and CCNB1/2 (dashed black borders in Fig 3). These genes were not included in the refined list of proliferation markers used for calculating pseudo-proliferation index because they did not consistently correlate with NCI60 growth rates, but they strongly correlate with relative cell proliferation when integrating more cell lines and more data sets. Although the selected set of proliferation markers used to calculate pseudo-proliferation index mainly consists of genes involved in DNA repair, many genes of the proliferation signature are involved in other parts of the cell division cycle. Fig 3 shows that many of them are involved in M-phase processes such as chromosome segregation. This indicates that our strategy for selecting signature genes was not biased towards S-phase functions but retrieved genes for which expression corelated with cell proliferation for reasons that are yet to determine.

## Use case 1: Proliferation signature in drug screens

Many drugs affect cell proliferation, thereby decreasing the proportion of cells actively dividing in samples. This can confuse data analysis when investigating drug mode of action because many of the genes regulated upon treatment are in fact correlated with cell proliferation. A recently published paper provides the proteomes of five cell lines after 53 drug treatments [7]. In many experiments, the proliferation signature was enriched for the proteins that were downregulated after treatment, suggesting that the drug treatments reduced cell proliferation rates.

After brefeldin A [26] treatment, 10% of the downregulated proteins ($q$-value $\leq 0.05$) were proliferation signature genes ($p$-value $< 10^{-15}$; Fig 4a). Brefeldin A disassembles the Golgi complex and induces endoplasmic reticulum (ER) stress. It is usually used as potent inhibitor of cell secretion. Consequently, Brefeldin A treatment reduces cell proliferation, which is very visible when labelling signature genes in the volcano plot Fig 4b (orange dots): most of them are shifted towards the left of the volcano. Labelling them facilitates data analysis by: 1) highlighting global fold-change shifts that can be due to proliferation increase or decrease as a consequence of drug treatment and 2) disregarding protein regulations due to proliferation changes if these are not the main focus of the experiment to concentrate on more direct consequences of drug treatment.

Docetaxel treatment impacts cell proliferation specifically in A549 cells (lung carcinoma epithelial cells) where 47% of the proteins significantly downregulated ($q$-value $\leq 0.05$) were signature genes ($p$-value $< 10^{-15}$, Fig 4c). The volcano plot corresponding to this experiment is presented Fig 4d. Docetaxel is a taxane that interferes with microtubule growth by binding to the β-subunit of tubulin. It is used in the treatment of many cancers. Fig 4e shows the STRING network of functional associations of the proteins significantly downregulated in Fig 4d (grey box). Most of these genes are functionally connected in a "hairball" that contains all but one signature gene. Some of these hits are involved in microtubule remodeling, but others are downregulated because of a reduction of cell proliferation of the A549 cells upon treatment. Examples of the latter include RRM2, which catalyzes the biosynthesis of deoxyribonucleotides, and the chromatin-assembly factors CHAF1A and CHAF1B. Labeling the proliferation signature facilitates the identification of proteins potentially more relevant to the drug treatment (grey nodes outside of the hairball). For example, the Microtubule-associated tumor suppressor 1 (ATIP3, coded by the gene *MTUS1*). *MTUS1*-deficiency is associated with increased microtubule dynamics [27], which is the opposite of docetaxel-induced microtubule stabilization. In breast cancer, ATIP3 was found significantly downregulated in taxane-sensitive tumors [28]. It is an interesting therapeutic target for breast cancer [29]. Caspase 2 (*CASP2*)

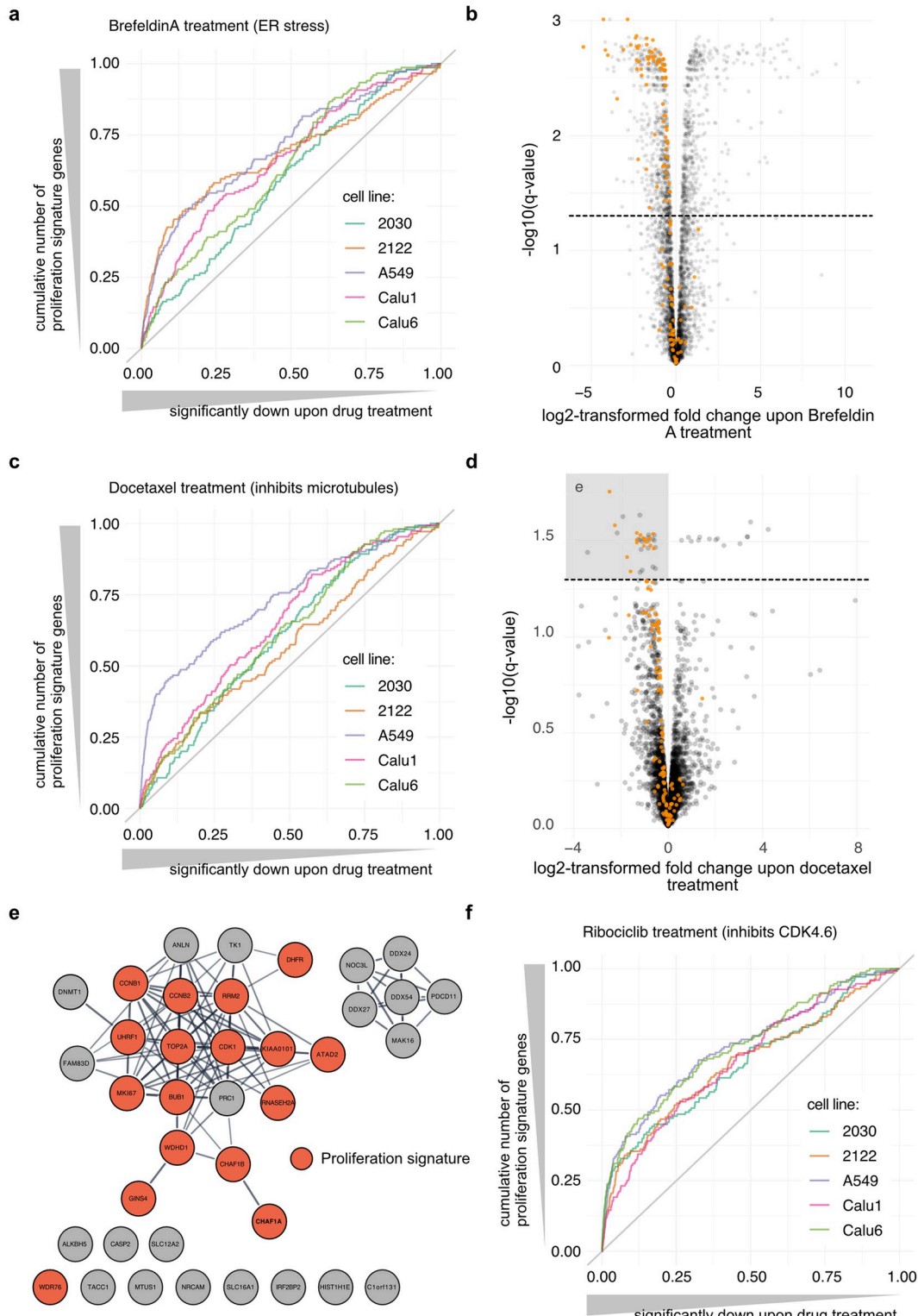

**Fig 4. Proliferation signature in the context of drug treatment. a)** Enrichment of the proliferation signature in the proteomes of cells treated with Brefeldin A. Proteins were ranked by significance of down-regulation according to Ruprecht *et al.* [7] (*q*-value) (horizontal axis) and the vertical axis presents the cumulative number of signature genes for each cell line. "2030" and "2122" correspond to the NCIH-2030 and NCIH-2122 cell lines, respectively. **b)** Volcano plot for A549 cells treated with Brefeldin A. Proliferation signature genes are highlighted in orange. The dashed line corresponds to a *q*-value of

0.05. **c)** Enrichment of the proliferation signature in the proteomes of cells treated with Docetaxel as in (a). **d)** Volcano plot for A549 cells treated with Docetaxel as presented in (b). **e)** Significantly down-regulated proteins (grey square in (d)) are presented in a STRING network of functional associations (score ≥ 0.7). Proliferation signature genes are highlighted in orange. **f)** Enrichment of the proliferation signature in the proteomes of cells treated with Ribociclib as in (a).

has been shown to cleave the Microtubule-associated protein tau (coded by the gene *MAPT*) that promotes microtubule assembly and stability and potentially competes with taxanes for microtubule binding. It is associated with resistance to taxanes in several cancers [30–32]. The Transforming acidic coiled-coil-containing protein 1 (TACC1) is also involved in microtubule regulation [33]. The Nucleolar complex protein 3 homolog (*NOC3L*), protein MAK16 homolog, RRP5 homolog (*PDCD11*) and the ATP-dependent RNA helicases DDX24/27/54 are RNA-binding proteins. Although there is no obvious known association of these proteins with docetaxel treatment and/or microtubule regulation, these downregulated proteins may inform on docetaxel impact on A549 cells.

In other cases, such as ribociclib treatment the same genes are not to be set aside but reflect the drug mode of action. Ribociclib inhibits CDK4/6 activity and thereby prevents progression through the G1/S checkpoint, blocking cells in G1 phase. This results in a high enrichment of the proliferation signature in negatively regulated genes (Fig 4f), which is highly relevant for data interpretation.

## Use case 2: Proliferation signature in the context of cancer prognostic and classification

The proliferation signature can also be useful for analysis of *in vivo* samples and patient data, for example in the context of cancer since most tumors are characterized by an increased proliferation rate. Many of the signature genes reported here are indeed reported prognostic markers in the context of cancer. This can be highly relevant since these genes may be significantly regulated because of the presence of more dividing cells in certain tumor samples. Other genes/proteins may be more appropriate for targeted therapy.

The recently published meta-analysis of the Clinical Proteomic Tumor Analysis Consortium (CPTAC) [6] identified proteins which relative quantities are correlated with tumor grade or stage in patient samples. The proliferation signature genes identified in this study were not enriched in proteins associated with tumor stage (Fig 5a). Proteins strongly correlating with tumor grade, however, were enriched in the proliferation signature in lung adenocarcinoma (LUAD), uterine endometrial carcinoma (UCEC), and pediatric glioma, but not in clear cell renal cell carcinoma (CCRCC) and ovarian serous adenocarcinoma (OV) (Fig 5b). This is in agreement with the GO-term enrichment presented in Monsivais *et al.* [6], where "cell cycle process" and "DNA replication" are strongly enriched in the proteins the most associated with cancer grades in LUAD, glioma and UCEC.

In the lung adenocarcinoma and pediatric glioma data, the proteins the most associated with cancer grade include a high number of signature genes of cell proliferation that may not be the best candidates for targeted treatment. Fig 5c shows the proteins correlation with grade (vertical axis), with signature genes highlighted in orange. With such figure, it is possible to quickly identify proteins that are specifically correlated with high tumor grade but not associated with the high proliferative state of aggressive lung tumors.

In lung adenocarcinoma, the three proteins the most correlated to cancer grade belonged to the proliferation signature: the U3 ubiquitin-protein ligase UHRF1, Kinesin-like protein KIF11 and the well-known proliferation marker MKI67 FHA domain-interacting nucleolar phosphoprotein. The Anillin actin binding protein (*ANLN*) was the top hit amongst non-

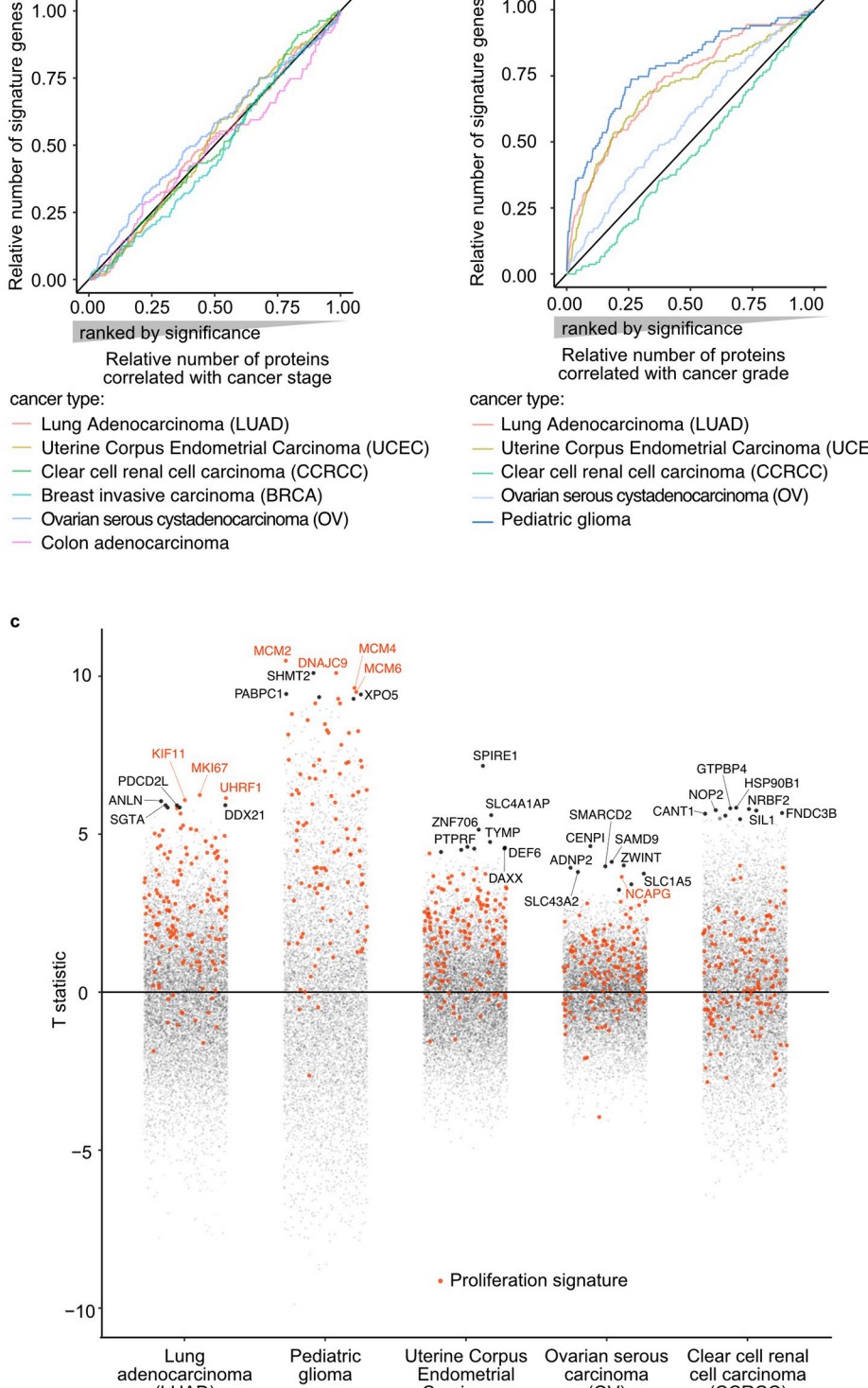

**Fig 5. Proliferation signature in the context of cancer grade. a-b)** Enrichment of the proliferation signature in proteins associated with cancer stage (a) and grade (b). Proteins were ranked by significance of correlation according to Monsivais *et al*. [6] (*p*-value of Pearson correlation) (horizontal axis) and the vertical axis presents the cumulative number of signature genes for each cancer type. **d)** Proteins T-statistic provided by Monsivais *et al*. [6] for the analysis of cancer grades (positive = high correlation with cancer grade) for each cancer type (horizontal axis). Each point

corresponds to a protein, signature genes are highlighted in orange. The seven top hits for each cancer type are indicated by their gene names.

signature genes. It is highly expressed in lung cancer cell lines and tumor samples compared to normal tissues [34], and *ANLN* high expression is a predictive marker of poor outcome for patients with lung adenocarcinoma in TCGA [35]. Anillin activates cellular migration of lung cancer cells *in* vitro [34] and increases tumor growth and metastasis in breast cancer through induction of mesenchymal to epithelial trans-differentiation [36].

76% of the signature genes identified in this study were correlated to cancer grade with a *p*-value under 0.01 in gliomas (Fig 5c). In such context, it is particularly important to acknowledge that these genes may be regulated because of differences in cell growth rates. The signature genes MCM2/4/6, and the heat shock co-chaperone and histone chaperone DNAJC9 [37] were amongst the five proteins the most correlated with glioma grades. In the figure, these surround the mitochondrial serine hydroxymethyltransferase (SHMT2) (ranked 3[rd]), which could be a more interesting hit for targeted therapy. It participates to the synthesis of glycine by catalyzing serine-to-glycine conversion. Glycine is a key resource for proliferative cells, and elevated concentration of glycine in IDH-mutated glioma tumors has been associated with aggressive glioma and is predictive of shortened patient survival [38]. While SHMT2 expression is not directly correlated with glycine concentration in gliomas [38,39] it has been shown to favor cancer cells adaptation to poorly vascularized tumor micro-environments in the context of ischemic glioma [39], and to be associated with poor prognostic in glioma [40].

Highlighting the proliferation signature in the analysis of LUAD and pediatric glioma allowed to quickly focus on proteins more directly associated with cancer grades in these contexts. This illustrates the advantage of taking these signature genes into consideration when analyzing proteomics data of patient samples.

## Conclusion

Here, we calculated a pseudo-proliferation index that we used as proxy for relative cell proliferation at transcript and protein level to define high-confidence thresholds for identifying a set of genes correlated with cell proliferation rates. We combined the transcriptomics and proteomics analysis to provide a final list of genes constituting a proliferation signature. The S3 Table provides the correlations to pseudo-proliferation index for 10,600 genes/proteins quantified in the data sets that were used for this analysis. With this list of signature genes, anybody can identify in their data sets the genes/proteins correlated with cell proliferation like contaminants are routinely flagged using the CRAPome [41].

We showed examples of high-throughput data analysis where labelling the proliferation signature facilitates data interpretation. It informs on the potential impact of differences in cell proliferation in a given experimental set up–which is for example strongly down-regulated upon treatment with drugs blocking the cell cycle–and can inform on differences in cell proliferation in tissue samples such as tumors.

We also show how cell growth rate can be a confounding factor that results in down- or up-regulation of many genes in *in vitro* drug screens and tumor samples. Flagging these confounders among the most regulated genes allows to quickly identify other regulated hits that could be more relevant in the context of the experiment. Such analyses still require strong knowledge of the biological context and molecular regulation at play, but the genes correlated with cell proliferation rates are not all annotated as being involved in replication of cell-cycle-related processes. Thus, our refined list of proliferation signature genes is an invaluable

resource for interpreting data where changes in cell growth rates/proliferation is a confounding factor.

The strategy that we describe here to identify the proliferation gene signature is straightforward and can be applied to many other types of confounding factors. The only requirement, which can be very limiting, is the availability of several high-dimensional data sets on samples where the confounding factor of interest can be quantified. We believe that taking such gene signatures into consideration should become part of the high-throughput data analysis routine and will facilitate data interpretation in many biological contexts.

## Materials and methods

### Retrieval and pre-processing of proteomics data

The raw data from Gholami *et al.* [10] were retrieved from the PRIDE proteomeXchange repository PXD005946 and searched against the Human reviewed protein database (download 12/03/2021 from Uniprot.org) with MaxQuant v1.6.17.0. The mqpar.xml and the fasta file associated with the search are available on Zenodo.org (10.5281/zenodo.6346643). The proteinGroups.txt table was filtered to remove the reverse sequences and potential contaminants identified with the contaminant database included in MaxQuant. We kept only the protein groups with minimum one unique peptide and a *q*-value ≤ 0.01 (6,900 protein groups). We further removed the samples with more than 70% of missing values. LFQ was utilized for correlation calculation after variance stabilizing normalization (vsn) [42].

The data from Guo *et al.* [11] were retrieved from the S1E Table provided in the paper (3,171 protein groups with no missing value) and normalized using vsn before correlation calculation.

The normalized iBAQ quantification from Frejno *et al.* [9] was retrieved from the supplementary Data 3 available with the paper. The tables for Trypsin, GluC and Trypsin digestion of the CRC65 cells were filtered to remove the reverse sequences and potential contaminants identified with the contaminant database included in the original MaxQuant search. We kept only the protein groups with minimum one unique peptide and a *q*-value ≤ 0.01 (9,744 and 7,271 protein groups in the trypsin and GluC dataset for the NCI60 cells, respectively and 11,308 for the CRC65 digested with trypsin). We further removed the protein groups with more than 50% of missing values. We also removed the protein "PLIB" in the trypsin dataset due to bad annotation. For the analysis of the NCI60 cell lines, we took the protein groups mean signal from the trypsin and the GluC data sets.

The normalized TMT quantification from Nusinow *et al.* [12] was retrieved from the supplementary data available on https://gygi.hms.harvard.edu/publications/ccle.html ("Protein Quantitation (TSV Format)"). The tables were filtered to remove the protein groups with more than 50% of missing values.

The data from Gonçalves *et al.* [21] were retrieved from the S1 Table provided in the paper (6,692 protein groups with a minimum of 2 quantified peptides) and normalized using median subtraction before correlation calculation. We removed the protein groups quantified in less than 10 cell lines (6,451 protein groups remaining), and the four following cell lines from the analysis because their names were too similar and could create mismatch between the different data sets: "TT", "T-T", "KMH-2" and "KM-H2".

### Proteome inter-data set matching

Since the searches were performed on each proteomics data set independently, the same protein can be labelled differently in the search outputs (*i.e.* belong to different protein groups, split across several isoforms. . .). We retrieved the protein groups corresponding to the same

protein in different data sets. We first combined variants/isoforms signal by keeping their mean values. Then, we matched and renamed them across data sets according to the mapping table that is provided as S1 Table. In the cases where several rows of a given data set were mapped to the same homogenized protein group ID, we kept the mean value per sample. If several accessions of a given data set corresponded to a unique accession in another data set, we favored the homogenized protein group ID with the highest number of matching protein groups across data sets. In cases of tie, we kept the one with the least "combined" accessions (several accessions corresponding to the homogenized accession in a given data set).

### Proteomics proliferation signature

For each data set independently, we calculated the mean of signal of proteins of the MCM complex (MCM2, MCM7, MCM3, MCM4, MCM5 and MCM6), CDK1, PCNA, PLK1, RPA2, RRM1, RRM2, RFC4, RFC2, FEN1, MKI67, PRIM2, POLA1, RPA1, RPA3, RFC5, RFC3, SMC1A, SMC3, STAG2 to generate a pseudo-proliferation index. The gene names and corresponding Uniprot accessions are provided in S4 Table.

For each protein group quantified in a minimum of 10 cell lines, we calculated its Pearson correlation to cell lines pseudo-proliferation index and to the growth rates calculated based on doubling time (available on dtp.cancer.gov/discovery_development/nci-60/cell_list.htm—Last Updated: 05/08/15). Missing values were replaced in each data set with the 1% quantile. We excluded the protein groups only quantified in one data set and calculated the mean of Pearson correlations. The absolute mean of Pearson correlation to pseudo-proliferation index was utilized to rank the protein groups. We performed the same analysis after randomization of the cell lines' pseudo-proliferation index (50 iterations); the distribution of the resulting absolute mean of Pearson correlations across data sets allowed us to define FDR thresholds: 0.1% FDR was obtained for an absolute mean of correlation to pseudo-proliferation index $\geq 0.189$ in the proteomics data. To define a confidence threshold, we benchmarked the list of proteins ranked by decreasing absolute Pearson correlation to pseudo-proliferation index with three gene lists of gold standards: B1 [23], B2 [19,24,25], and the periodic genes described in Cyclebase 3.0 [16]. Their cumulative count in the ranked list of proteins was utilized to select Pearson correlation values corresponding to high enrichment of gold standards.

### Retrieval and processing of transcriptomics data

The processed data for the Affymetrix NCI60 dataset (Pfister *et al.*) [14] was obtained from the GEO NCBI portal using the GeoQuery R package (v.2.60.0) [43]. Probesets of the HGU133-Plus2 chip were mapped to Ensembl genes using the custom annotation provided by BrainArray [44]. The mapping file for probesets to Ensembl transcripts was obtained from the BrainArray version 25 download page (brainarray.mbni.med.umich.edu), and Ensembl transcript-gene mapping was retrieved using the R package biomaRt [45] v2.48.3. Probeset intensities were averaged across replicates of the same cell line. Only gene-specific probesets were considered, probesets mapping to multiple genes were excluded. When multiple probesets corresponded to the same gene, the one with the highest mean signal across all cell lines was selected to represent the gene. In total, 16,608 Ensembl genes (of which 15,541 having the biotype "protein coding genes") were uniquely mapped to probesets on the chip for the 59 NCI60 cell lines.

Raw fastq files corresponding to the NCI60 RNA-Seq profiling (Reinhold *et al.*) [15] were obtained from the European Nucleotide Archive (project accession PRJNA433861). The raw sequence reads were trimmed using Trimmomatic v038 [46], using the adapter file "TruSeq3-PE-2.fa", and with the following parameters: "LEADING:3 TRAILING:3 SLIDINGWINDOW:4:15

MINLEN:36". Transcript abundance estimates were then obtained using salmon v1.4.0 [47] in "quant" mode with the default parameters against the Human GRCh38 cDNA set obtained from the Ensembl release 103 [48]. Gene-level abundance estimates were summarized using the R package tximport v.1.20.0 [49], and upper quartile normalization was performed with the calc-NormFactors function from the edgeR package v. 3.34.1 [50]. Finally, expression levels were obtained for 57,937 Ensembl genes (21,391 having the biotype "protein coding genes") for the same 59 cell lines profiled in Pfister *et al.*.

For the CCLE dataset (Ghandi *et al.*) [13], normalized gene expression levels in TPM (transcripts per million) units were obtained from the DepMap portal (depmap.org/portal, "CCLE_expression_full.csv"). We used the original Ensembl gene identifiers provided in the files: 51,832 Ensembl genes (19,790 protein coding) across 1,026 cell lines.

## Transcriptomics proliferation signature

For each dataset independently, the pseudo-proliferation index was obtained as described for the proteomics datasets, by averaging the expression levels of the selected proliferation markers. For each gene, in each dataset we computed the correlations with the pseudo-proliferation index calculated for the dataset, as well as correlation with growth rates using the NCI60 cell lines doubling times when available. We selected the genes quantified in at least two datasets and calculated the mean of Pearson correlations. We performed the same analysis after randomization of the cell lines' pseudo-proliferation index (50 iterations) to define FDR thresholds: 0.1% FDR was obtained for an absolute mean of correlation to pseudo-proliferation index $\geq 0.251$ in the transcriptomics data.

Mapping between Ensembl gene identifiers and UniprotKB swissport accessions has been performed using biomaRt [45]. For the integration of the RNA data with the proteome, we removed the genes from the transcriptome that matched to more than 6 protein groups in the proteome (2 genes) and reported the values of each gene from the transcriptome if they matched the same protein group (25 genes).

## Gene set enrichment and GO term redundancy reduction

Gene set enrichments were performed with R v4.0.3 (R-project.org/) and RStudio v1.3.1093 (rstudio.com/) on a x86_64-apple-darwin17.0 (64-bit) running macOS Big Sur 10.16, using the packages clusterProfiler v3.18.1 [51] and org.Hs.eg.db v 3.12.0. The protein accessions were ordered by decreasing Pearson correlation with growth rates or proliferation index. We ran the function gseGO() with the following parameters: ont = "ALL", keyType = "UNIPROT", minGSSize = 6, maxGSSize = 800, pvalueCutoff = 0.05, verbose = TRUE, OrgDb = "org.Hs.eg.db", eps = 0, pAdjustMethod = "BH". The output summary was used to make the S4 Fig that presents GSEA on data sets with only NCI60 cell lines (first 2 panels) or without any NCI60 cell (last panel). We then simplified the output to reduce GO terms redundancy globally: we calculated the pairwise Jaccard indexes between all pairs of GO terms identified across data sets. Pairs of GO terms with a Jaccard index $\geq 0.5$ were considered similar and only the one with the lowest enrichment *q*-value in any data set was kept for plotting. S4 Fig shows the 80 biological processes with the lowest absolute *q*-value (minimum value across all data sets and enrichments).

## Functional annotations and networks

The two gene/protein networks presented in this paper were generated with Cytoscape v 3.9.1 [52]. GO term annotations were retrieved with the StringApp v 1.7.0 [53] and the donut visualization of Pearson correlations was performed with Omics Visualizer v 1.3.0 [54].

## Proliferation signature in the context of drug treatment

The proteomics analyses of drug-treated cells were found in the supplementary Data 1 of Ruprecht *et al.* [7]. We counted the cumulative number of proteins subjected to statistical analysis by the authors and with a negative fold change upon drug treatment with Ribociclib (10,000 nM in all cell lines), Brefeldin A (100 nM, 30 nM, 100 nM, 100 nM, 30 nM for NHI-2030, NHI-2122, A549, Calu1 and Calu6, respectively) and Docetaxel (30 nM, 3 nM, 1 nM, 10 nM, 3 nM for NHI-2030, NHI-2122, A549, Calu1 and Calu6, respectively). Volcano plots were drawn with the data from Ruprecht *et al.* [7], proliferation signature genes were mapped to protein groups if minimum one of the proteins in the protein groups had a gene name corresponding to a signature gene.

## Proliferation signature in the context of cancer tissue samples

The proteomics analyses of tumor tissues were found in Montsivais *et al.* [6] (Supplementary Data 2 and 3 for correlation with grade and stages, respectively). Proliferation signature genes were mapped to protein groups if minimum one of the proteins in the protein groups had a gene name corresponding to a signature gene.

## Supporting information

**S1 Fig. Selection of proliferation markers for calculating pseudo-proliferation index in proteomics and transcriptomics data.** Volcano plots showing the mean correlation of proteins (a) or transcripts (b) to growth rates in the NCI60 data sets (horizontal axis) and the -log10(coefficient of variance) across all the data sets (vertical axis). Proteins quantified in less than 3 data sets were excluded in (a). Proteins/genes of interest are highlighted, and proliferation markers identified from literature search are indicated with triangles. These were color coded based of their expression peak according to Santos *et al.* [17].
(TIFF)

**S2 Fig. Robustness of pseudo-proliferation index with different sets of proliferation markers.** Pearson correlations between pseudo-proliferation index and growth rates in the proteomics data sets containing NCI60 cells presented in (Fig 1a) using the median (left panel) or mean (right panel) signal of the three sets of proliferation markers as selected in (Fig 1b) (grey area), all the previously reported proliferation markers, or the previously reported proliferation markers and cycling genes with the exclusion of RAD21. Grey points and bars are mean and confidence intervals across data sets. The right panel is the same as Fig 1c, it is reported here for direct comparison with the left panel.
(TIFF)

**S3 Fig. Cell lines in the different proteomics data sets.** Number of cell lines in the proteomics data sets used in the study. The total number of cell lines in the data sets are indicated in the left-hand side bar plot (color-coded by the cell line panel). The cell lines present in multiple data sets are indicated by the bar plot on the top: number of protein groups detected in the data sets indicated by a dot on the dot plot. Each data set is identified by the first author's name.
(TIFF)

**S4 Fig. Comparison of the biological functions enriched in proteins strongly correlated with cell lines growth rates or pseudo-proliferation index.** For each data set, genes (with the exception of the genes used for calculating pseudo-proliferation index) were ranked based on their correlation to growth rates or correlation to pseudo-proliferation index (left and right

panel, respectively). Gene set enrichments were performed using the "gseGO" function from the R package clusterProfiler v 3.18.1, resulting *p*-values are indicated in each tile, as well as color-coded normalized enrichment scores (NES). Only the annotations from biological processes are included, they are ordered by decreasing maximum NES per data set (top 80 enriched GO terms, see material and methods for a detailed description of the procedure used to reduce GO redundancy). Data sets are labeled based on the first author's name, enrichments were performed independently on the NCI60 cell lines or cell lines with no reported doubling time ("NCI60 only" and "NCI60 excluded", respectively).
(TIFF)

**S5 Fig. Proteomics data and Brefeldin A treatment. a)** Protein coverage of the proteomics data sets used in the study (after isoform removal and accessions homogenization—see material and methods). These are identified by the first author's name (left). The data set "Frejno" contained two independent MS searches of different cell line panels, we kept them separated. The total number of protein groups detected in the data sets are indicated in the bar plot on the right-hand side (color-coded by the cell line panel). The protein groups identified in multiple data sets are indicated by the bar plot on the bottom: number of protein groups detected in the data sets indicated by a dot on the dot plot. **b)** Volcano plots for each cell line treated with Brefeldin A. Genes constituting the proliferation signature are highlighted in orange. The dashed line corresponds to a *q*-value of 0.05.
(TIFF)

**S1 Table. Accession mapping between the proteomics data sets.** The proteomics data sets did not all group the proteins the same way (due to differences in peptide coverage). This table indicate which groups were mapped to which accession in this study.
(XLSX)

**S2 Table. Pseudo-proliferation indexes calculated in each data set.** Pseudo-proliferation indexes calculated for each proteomics and transcriptomics data set.
(XLSX)

**S3 Table. Gene/protein correlations to pseudo-proliferation indexes.** Gene and protein Pearson correlation to pseudo-proliferation indexes in each data set, mean across data sets and associated FDR. This table contains the final list of signature genes (TRUE values in the column "Signature gene").
(XLSX)

**S4 Table. Genes/proteins used for calculating pseudo-proliferation index.** List of genes and protein accessions selected for calculating the pseudo-proliferation index.
(XLSX)

## Author Contributions

**Conceptualization:** Marie Locard-Paulet, Oana Palasca.

**Formal analysis:** Marie Locard-Paulet, Oana Palasca.

**Funding acquisition:** Lars Juhl Jensen.

**Investigation:** Marie Locard-Paulet, Oana Palasca.

**Methodology:** Marie Locard-Paulet, Oana Palasca, Lars Juhl Jensen.

**Supervision:** Lars Juhl Jensen.

**Visualization:** Marie Locard-Paulet.

**Writing – original draft:** Marie Locard-Paulet.

**Writing – review & editing:** Marie Locard-Paulet, Oana Palasca, Lars Juhl Jensen.

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
