## [Decision Letter · Decision Letter 0]

17 Jul 2022

Dear Dr Locard-Paulet,

Thank you very much for submitting your manuscript "Identifying the genes impacted by cell proliferation in proteomics and transcriptomics studies" for consideration at PLOS Computational Biology.

As with all papers reviewed by the journal, your manuscript was reviewed by members of the editorial board and by several independent reviewers. In light of the reviews (below this email), we would like to invite the resubmission of a significantly-revised version that takes into account the reviewers' comments.

Please consider the suggestion by Reviewer 1 and address the comments raised by all three referees.

We cannot make any decision about publication until we have seen the revised manuscript and your response to the reviewers' comments. Your revised manuscript is also likely to be sent to reviewers for further evaluation.

Sincerely,

Attila Csikász-Nagy

Associate Editor

PLOS Computational Biology

William Noble

Deputy Editor

PLOS Computational Biology

Reviewer's Responses to Questions

**Comments to the Authors:**

Reviewer #1: The manuscript by Locard-Paulet et al. describes a compilation and analysis of transcriptomic and proteomic datasets to identify genes whose expression is highly correlated with cell proliferation rates. The study is straightforward. The authors queried all the available datasets and identified gene expression signatures of cell proliferation with high confidence. In a clever extension, they also used these markers to predict proliferation rates in cases/cell lines for which cell proliferation rates are unknown. Overall, the work is solid, and the datasets will be of value to researchers. However, the way the authors pitch their findings is unfortunate and, in my opinion, wrong. I do not understand why the authors recklessly undermine the significance of their work. For the reasons I will detail below, I think the manuscript ought to be re-written before publication in any journal.

MAJOR CONCERNS

1. The authors present their findings throughout the manuscript as if the genes they identified are garbage and irrelevant. They call them confounders and envision their work analogous to the CRAPome reported previously for spurious protein-protein interactions. I cannot fathom the reasons for this genuinely bizarre interpretation. Growth rate as a variable is of immense significance in itself. Innumerable omic screens are done with cell proliferation rates as the critical phenotypic output (probably most cancer-related, cell-based screens). Their own 'use case 2' makes this point, where many of their 'confounders' are validated targets implicated in cancer. For example, the earlier work they cite from just one dataset (ref 1) identified the Myc/E2F network as a major contributor to proliferation rates, which is highly physiologically relevant. There is nothing 'confounding' or 'CRAPome’-related. The authors did some nice analyses, querying multiple datasets, and coming up with gene lists that will undoubtedly help others. They should present their work as follows: First, describe their goal exactly as it is: Identify gene expression signatures associated with proliferation rates. Whether such signatures are confounders or CRAPome-related depends on each screen's purpose. Then, explain that for cases where growth rate is the point of interest, users will have a gold standard reference at their disposal. If the growth rate is not of interest (similar to their 'use case 1') as in screens looking for a drug target, their list will help eliminate indirect effects due to growth rate changes. I understand that this would require a significant re-editing of the whole manuscript, but it would cast the work in a more positive, inclusive context.

2. I find their development of the pseudo-proliferation index very useful. However, they offer no experimental validation. The authors should measure the doubling times of some cell lines for which no data is available and see if they match their computational estimates. This is a simple experiment that should be done to test their predictions.

MINOR

Line 229: Small typo. Change Befeldin to B*r*efeldin.

Reviewer #2: Review of “Identifying the genes impacted by cell proliferation in proteomics and transcriptomics studies” by Locard-Paulet, Palasca and Juhl Jensen.

In the work presented in this manuscript, Locard-Paulet and co-authors revisit recently published transcriptomics and proteomics papers comparing large numbers of cell lines, looking across all cell lines and datasets for genes (or proteins) whose expression (or abundance) correlate with cell proliferation and therefore possibly confounding the interpretation of experiments whose design did not adequately take differential cell proliferation or proportion of cells at different cell cycle phases into account, for example when comparing treated cells with non-treated controls. Although many researchers are undoubtedly aware of this, and correctly identify such effects from their data (e.g. by looking at enriched biological processes corresponding to DNA replication or cell division - however incomplete the gene annotation), the current manuscript tells a compelling story that will serve to increase awareness of cell cycle confounders and provide a practical index based on a set of proteins strongly and consistently affected by changes in cell proliferation for estimating relative cell proliferation in proteomics datasets without information on cell growth rates. The authors also demonstrate how the index and confounders can be used in the context of cancer drug treatments, where the drug may directly affect the abundance of one or more of the confounders, showing that this small set of confounders can be used as a metaphorical lens through which one can look at different types of data.

The manuscript addresses an important problem and is very well written. The conclusions are supported by the data. All relevant details on the methods, underlying data and code are available as supplemental information and on Zenodo, enabling others to use and modify this pseudo-proliferation index or adapt it for other purposes. I could access these files and open the Cytoscape files without issues.

However, I do have only a few questions to the authors, and some minor comments and suggestions:

Robustness of the index:

How robust is the pseudo-proliferation index? It is calculated as a mean of the signal of 22 proteins, making it sensitive to an outlier that could be caused by a single false peptide identification or interfering species. Would it not be better to use the median, or some other more robust statistic?

Will counting all MCM proteins individually not give too much weight to the MCM complex?

Applicability of the index:

The genes/proteins selected for the pseudo-proliferation index calculation include only those previously functionally annotated with “DNA replication”, and none of those annotated with “chromosome segregation” (though RAD21 and MCM4 are apparently known to peak in the M-phase). The genes/proteins in the index were those with the highest correlation with known growth rates in the data used. How representative is this set for all types of confounding issues related to cell proliferation, in other cell lines or model organisms? Perhaps the authors could comment on this in the discussion?

Minor comments/suggestions:

The authors correlate gene and protein abundance with the inverse of the cell doubling times, and use both “growth rate” and “inverse doubling time” semi-interchangeably in the manuscript. Would it make it easier for the reader to simply use “growth rate” [ln(2)/doubling time] throughout the text and in the figures, rather than both “growth rate” and “1/doubling time”?. The change should not alter the results or conclusions in any way.

Figure 1b: What is meant by “coefficient of variance in the proteomics”? There seems to be a word missing at the end of this sentence, e.g. “data”. What intervals do the three sizes or the markers represent?

Figure 1, 2, 4 and 5: There are strictly speaking no x- or y-axis defined in these plots, though most readers would understand “x-axis” to mean the horizontal axis (abscissa) and “y-axis” the vertical (ordinate) axis.

Reviewer #3: Locard-Paulet et al. focus on the systematic identification and quantification of cell proliferation effect in high-throughput transcriptomic and proteomics studies and its role as potential confounder in differential analyses. The authors set out to explore this effect using large scale cohorts of cell lines and identified 223 genes that are strongly associated with proliferating rates, expanding existent resources. Several examples and two use cases using both tumor and drug perturbation studies are presented to support the role of cell proliferation as a major factor in some contexts.

This analysis is aimed at assisting large scale hypothesis exploration studies to account for the confounding effects of growth rates in the identification of spurious associations. The computational and statistical analyses are well executed and the manuscript is clearly written. In general, this is an important question, nonetheless cell proliferation does not seem to affect baseline transcriptomics and proteomics extensively, and drawing the line between confounder and a biologically relevant feature is sometimes difficult. Hence, the standard application of cell proliferation as a covariate in large-scale association studies should not be recommended without prior evaluation. In fact, as already explored in cancer cell lines studies, cell proliferation impact is more pronounced in short term perturbation studies, such as drug response assays, as the authors confirm in the first use case, where faster growing cells show on average stronger drug responses.

Main comments:

1. It would be important to complement these analyses with more up-to-date and systematic information of cancer cell lines proliferation rates. This can be found in these couple of publications PMID:31068703 (doubling times for 708 cell lines in Supplementary Table 1) and PMID:35839778 (growth rates for 938 cell lines in Supplementary Table 1).

2. Some general statistics about the preponderance of cell proliferation could be informative to assess the importance of this in the datasets at hand. For example, using Principal Component Analysis (PCA) is the proliferation gene set associated with any principal component in specific? If so, how much variability does it explain?

3. Page 9, line 195: “This indicates post-translational adjustment of protein quantities.”. This difference could also be attributed to post-transcriptional regulatory processes, correct?

Minor comment:

1. Page 3, line 82: “sixteen were quantified in minimum three of the NCI60 (...)”. Perhaps, “(...) in at least three (...)”?

**Have the authors made all data and (if applicable) computational code underlying the findings in their manuscript fully available?**

Reviewer #1: Yes

Reviewer #2: Yes

Reviewer #3: Yes

PLOS authors have the option to publish the peer review history of their article (what does this mean?). If published, this will include your full peer review and any attached files.

Reviewer #1: No

Reviewer #2: No

Reviewer #3: No
---

## [Decision Letter · Decision Letter 1]

26 Sep 2022

Dear Dr Locard-Paulet,

We are pleased to inform you that your manuscript 'Identifying the genes impacted by cell proliferation in proteomics and transcriptomics studies' has been provisionally accepted for publication in PLOS Computational Biology.

Best regards,

Attila Csikász-Nagy

Academic Editor

PLOS Computational Biology

William Noble

Section Editor

PLOS Computational Biology

Reviewer's Responses to Questions

**Comments to the Authors:**

Reviewer #1: Nice job revising the paper.

Reviewer #2: Review of revised manuscript "Identifying the genes impacted by cell proliferation in proteomics and transcriptomics studies" by Locard-Paulet and co-workers.

The authors have answered the questions I had, and made the necessary changes. I agree with the authors explanation for why the MCM proteins are counted individually. Perhaps in the future, the weight assigned to different proteins in the calculation of the index could be revisited and optimized.

The authors do not want to speculate about the applicability of the proliferation index to other cell lines and non-human organisms (e.g. model systems). This is fair enough. I agree with what the authors wrote in their response that the index is based on highly conserved cell cycle-regulated genes, and that it therefore could be expected to work also for other eukaryotes. But of course this still needs to be shown.

Reviewer #3: The authors should be commended for their work addressing and integrating the comments. I am satisfied with their answers and I believe this is an interesting study that will be of interest to the community.

**Have the authors made all data and (if applicable) computational code underlying the findings in their manuscript fully available?**

Reviewer #1: Yes

Reviewer #2: Yes

Reviewer #3: Yes

PLOS authors have the option to publish the peer review history of their article (what does this mean?). If published, this will include your full peer review and any attached files.

Reviewer #1: No

Reviewer #2: No

Reviewer #3: No

---

## [Editor Report · Acceptance letter]

30 Sep 2022

PCOMPBIOL-D-22-00916R1 

Identifying the genes impacted by cell proliferation in proteomics and transcriptomics studies

Dear Dr Locard-Paulet,

I am pleased to inform you that your manuscript has been formally accepted for publication in PLOS Computational Biology. Your manuscript is now with our production department and you will be notified of the publication date in due course.

With kind regards,

Zsofia Freund
